# Investigation into Micro-Polishing Photonic Crystal Fibers for Surface Plasmon Resonance Sensing

**Qingmin Liu, Junpeng Chen, Shanglin Hou * and Jingli Lei**

School of Science, Lanzhou University of Technology, Lanzhou 730050, China
* Correspondence: houshanglin@vip.163.com; Tel.:+86-13893697556

**Abstract:** In this work, we propose and demonstrate a micro-polishing-fiber (MPF)-based surface plasmon resonance (SPR) sensor. The structure of the sensor is simple and consists of three layers of regular air holes and two small air holes. The sensitivity seldom depends on the sizes of the air holes, which leads to a sensor with high structure tolerance. A tiny polishing depth ensures the mechanical strength of the polished fiber. There are three decisive factors for mass production and application of the sensor. A thin layer of indium tin oxide (ITO) film is applied to the polished surface to excite plasmonic interactions and facilitate refractive index (RI) detection. The SPR sensor is designed and analyzed by the finite element method (FEM), and optimized in terms of the air holes' diameter, the ITO film thickness, and the core-to-surface interval. In the wide detection range between 1.32 and 1.39, the wavelength sensitivity can reach up to 11,600 nm/RIU. The MPF–SPR sensor exhibits great potential in the fields of optics, biomedicine, and chemistry.

**Keywords:** micro-polishing; photonic crystal fiber; sensor; surface plasmon resonance

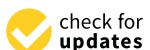



## 1. Introduction

Surface plasmon resonance (SPR) is a common optical phenomenon referring to the surface plasmon polariton (SPP), which is formed when light waves occur at the metal and dielectric interface. Surface plasmon wave (SPW) excites electron density oscillations at the metal–dielectric interface. When SPR occurs, the effective RI of SPP mode is extremely sensitive to variations in the RI of the material on the metal surface, and the RI of analyte can be reflected by detecting the spectral or power characteristics of the transmitted light. At the same time, with the increasing demand for the detection of RI in the fields of biomedicine, biochemistry, materials science, nano-photonics, and environmental monitoring, SPR has been widely used for sensing [1–5]. The fundamental principle of SPR sensors is that the core-guided light can couple to the SPP on the metal surface during the phases of core-guided mode and SPP mode match. A coupling method such as prism or grating coupling should be used to excite SPW. Among them, the prism configuration has shown considerable sensitivity to SPR systems, but has disadvantages including the sensor's large size, high cost, and poor reliability [6]. Although the fiber-based SPR sensors offer a smaller structure than a prism SPR sensor, they have low sensitivity owing to a small overlap with their surroundings and low single-structure sensitivity [7]. These defects increasingly hinder their application in sensing and mass production.

The appearance of photonic crystal fiber (PCF) changes this dilemma and brings new development opportunities for SPR sensing. PCF is receiving considerable attention from researchers because of its micro volume, flexible structure, and excellent transmission characteristics [8–10]. Porous cladding is the unique structure of PCF; we can arrange air holes flexibly within the cladding around the fiber core. By carefully adjusting the structural parameters of PCF, such as the metal film thickness, arrangement of holes, hole diameter, and hole spacing, part of the light energy is confined to the core and a large part of the light field propagates in the air holes as an evanescent field. This means that the

mode field overlap of PCF is much larger than that of ordinary fiber and the light field in the cladding region is easier to control. What is more, the SPR sensor with higher sensitivity can be obtained.

Based on stimulating the resonance between the SPP mode (metal/silica or metal/liquid) and core mode in PCF, the plasmonic technology offers smaller, highly integrable, low-cost, real-time, high-sensitivity optical biosensors. Most SPR–PCFs proposed for the detection of RI of analyte can be classified into three categories, including tiny air holes' inner coating, fiber outer wall coating, and coating on the surface after polishing the fiber. Micro pores are introduced into the fiber core and metallic films are deposited into them to support SPR [11]. With a larger air–filling ratio and smaller space, the resonance effect is enhanced. One serious downside of microporous internal coating is the difficulty in the fabrication of depositing the metal film into micro inner walls [12]. Furthermore, the experimental results show that, using this method, it is not only difficult to fill and remove measured gas or liquid samples from the micro pores, but also real-time detection is hindered [13]. A PCF–SPR sensor with a fiber outer wall coating is dual-polarized highly sensitive, so exchanging the measuring sample will be simple [14]. It is also hard to maintain uniform thickness and roughness of the metal film for a fiber outer wall coating with the existing PCF manufacturing technologies [15]. To solve these crucial problems, PCF–SPR sensors that metallize on the flat surface after removing part of the cladding are proposed, including D-shaped fibers and U-shaped fibers [10,15,16]. However, for the D-shaped fiber, almost half of the cladding is polished off in the production process. The fiber is often close to being fractured because of its vulnerability and the mechanical strength is greatly reduced. The U-shaped fiber also requires a deep polishing depth, which makes it extremely hard to guarantee the completeness of air holes. The high grinding difficulty is not conducive to mass production. To make the PCF–SPR sensor more conducive to practical application by reducing the polishing depth, Chen et al. [17] proposed a micro-polishing PCF consisting of two hexagonal rings and an open-ring air hole channel. The simulation results show that their sensor can detect RI ranging from 1.20 to 1.29, and the maximum wavelength sensitivity is 11,055 nm/RIU. Because the light is confined in the PCF waveguide structure, the materials of the core, metal film, and cladding have an influence on the properties of the PCF–SPR sensors. Recently, many scholars have used cheaper ITO as the sensitive materials instead of the usual gold or silver [18,19]. This causes the evanescent wave to penetrate the deeper analyte and improves the coupling effects between the SPP mode and core-guided mode.

In this paper, an ITO-coated micro-polishing-fiber (MPF)-based SPR sensor is proposed, consisting of three layers of regular air holes and two small symmetrical air holes. The PCF–SPR sensors are usually characterized by the wavelength sensitivity and the sharpness of loss peaks, and we take them as optimization goals. In the simulations, FEM is used to calculate the MPF–SPR sensor. We systemically investigate the diameter of the air holes, the thickness of the ITO film, and the core-to-surface interval for analyzing influences of different structural parameters on the sensing performance.

## 2. Geometric Structure and Numerical Modelling

The schematic of the proposed sensor is shown in Figure 1a. It comprises three layers of regular air holes and two small symmetrical air holes in an MPF–SPR sensor structure with no mid-upper air hole. The unique eccentric core structure is formed between two types of air holes, and the arrangement of regular air holes is rotationally symmetric in an equilateral triangle or a square [20,21]. These holes are introduced to lower the average RI of the edge and limit the light energy in the eccentric core. The two small air holes directly above the eccentric core are used to set up a light leakage channel to adjust the intensity of the evanescent wave that interacts with the SPP modes [22]. The vertical channel makes the evanescent field of eccentric core more likely to leak into the ITO film sensing area directly above the core. The critical factor to enhance the sensing performance of a PCF–SPR sensor is to improve the proportion of the evanescent field, thereby strengthening the interaction

of the evanescent field with the external analyte. After a small section of PCF is polished, a uniform thin ITO film layer is coated in the base of the flat surface for SPR excitation. The selective filling of samples is easy to exchange in the outer spacious circular channel and ensure real-time measurement.

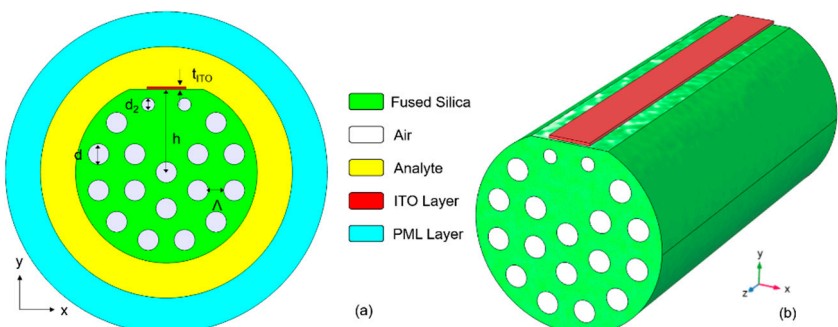

**Figure 1.** (**a**) Cross section of the proposed MPF–SPR sensor and (**b**) its three-dimensional view.

Figure 1b shows the three-dimensional view of the MPF–SPR sensor. The PCF preform can be fabricated using the state-of the-art technique of stack-and-draw [23,24]. During the fabrication process, the temperature, speed, and the stability of the drawing tower need to be well controlled to obtain satisfactory air holes. In order to achieve a micro-polishing shape, the wheel polishing technique can be used to side-polish the fiber to remove the top edge. Finally, a thin ITO layer can be uniformly coated on the flat sensing surface with accurate chemical vapor deposition [25] or magnetron sputtering [26]. The radius of the whole MPF–SPR sensor is set as 7 μm. The thickness of the ITO film layer is 60 nm, and the distance between the fiber core and the bottom of the ITO layer h is 6.4 μm. The diameters of large and small air holes are d = 1.6 μm and $d_2$ = 1 μm, respectively, and the distance between the two holes is (pitch) Λ = 2.8 μm.

We numerically investigate the sensing performance of this sensor using FEM with perfect matching layer (PML) and scattering boundary conditions [27]. The method can reduce the effect of reflection on the calculation results to mimic the real situation. In the selection of commercially available software, COMSOL Multiphysics simulate the transmission characteristics of the guided modes in the wavelengths range of 1400–2100 nm to find the effective refractive indices. In the structure, the RI of air is 1 and the cladding material is pure silica, whose wavelength dependence of the RI is calculated by the following Sellmeier equation [12]:

$$n(\lambda) = \sqrt{1 + \frac{B_1{}^2}{\lambda^2 - C_1} + \frac{B_2{}^2}{\lambda^2 - C_2} + \frac{B_3{}^2}{\lambda^2 - C_3}} \tag{1}$$

In Equation (1), coefficient $B_1$ = 0.696166300, $B_2$ = 0.407942600, $B_3$ = 0.897479400, $C_1$ = 4.67914826 × $10^{-3}$ μm², $C_2$ = 1.35120631 × $10^{-2}$ μm², $C_3$ = 97.9340025 μm², and λ represents the incident light wavelength in vacuum. Meanwhile, the thin ITO film is used as the SPR activity material, whose material dispersion is calculated by the Drude model:

$$\varepsilon_{\mathrm{m}}(\lambda) = \varepsilon_\infty - \frac{\lambda^2 \lambda_{\mathrm{c}}}{\lambda_{\mathrm{p}}^2 (\lambda_{\mathrm{c}} + i\lambda)} \tag{2}$$

In this expression, $\varepsilon_\infty$ = 3.8 is the infinite frequency dielectric function of ITO, while $\lambda_{\mathrm{p}}$ = 5.6497 × $10^{-7}$ m and $\lambda_{\mathrm{c}}$ = 11.21076 × $10^{-6}$ m are the plasmonic and collision wavelengths of ITO. The confinement loss of the fiber transmission modes is used to evaluate the properties of the SPR sensor and is obtained as follows:

$$\alpha_{\mathrm{loss}} = 8.686 \times \frac{2\pi}{\lambda} \mathrm{Im}[n_{\mathrm{eff}}] \times 10^6 \, (\mathrm{dB/m}) \tag{3}$$

where Im[$n_{eff}$] stands for the imaginary part of the effective RI. The wavelength sensitivity and the sharpness of loss peaks are commonly used as significant parameters to evaluate the performance of SPR sensors. According to the wavelength interrogation method, the wavelength sensitivity can be defined as follows:

$$S_\lambda(\lambda) = \frac{\Delta\lambda_{peak}}{\Delta n_a} (\text{nm/RIU}) \tag{4}$$

where $\Delta\lambda_{peak}$ denotes the variation in the resonant wavelength and $\Delta n_a$ is the analyte RI difference. The sharp loss peak is clear enough to detect the maximum point of loss spectra for the analyte. It has a small width half maximum (FWHM) that can control and filtrate the spectral noise.

## 3. Sensing Characteristics and Performance Analysis

### 3.1. Transmission Characteristics of the PCF Sensors

In any SPR-based sensor, when the phase matching condition is attained, the SPP mode created through the core-guided light can couple to the metal surface. Figure 2 illustrates the confinement loss and mode effective RI versus the operating wavelength of the core-guided mode and SPP mode. For our MPF–SPR sensor, two kinds of polarized core modes and the SPP mode can be obtained in the orthogonal direction. The imaginary part of their effective RI can be used to calculate the confinement loss and dispersion can be characterized by the real part. The confinement loss peak means that resonance coupling occurs, and the y-polarized core mode is coupled to the SPP mode. At the resonant wavelength around 1724 nm, the real parts of the effective RI of the y-polarized core mode and the SPP mode are equal. In that case, the loss of the y-polarized core mode reaches a maximum and the loss of the SPP mode reaches a minimum; they are equal because the complete coupling occurs [28]. Obviously, the confinement loss of the x-polarized core mode is not high enough to be detected, and the sensing characteristics of the y-polarized core mode are analyzed in this paper.

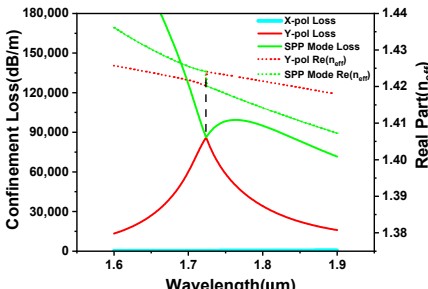

**Figure 2.** The confinement loss and dispersion relations of the core mode and SPP mode of the MPF–SPR sensor. The RI of the analyte is $n_a$ = 1.38.

Figure 3 displays light energy flow distributions of the y-polarized core mode and SPP mode for different wavelengths. At λ = 1600 nm (a shorter wavelength with respect to the resonant wavelength), most of the light energy is confined to the eccentric core, and only a small amount of energy leaks into the sensing surface. It is far away from the best phase matching and the confinement loss is low. At λ = 1800 nm (a longer wavelength with respect to the resonant wavelength), part of the energy leaks to the sensing interface. It is obvious that the maximum energy transfers from the y-polarized core mode to the SPP mode at λ = 1724 nm (resonant wavelength).

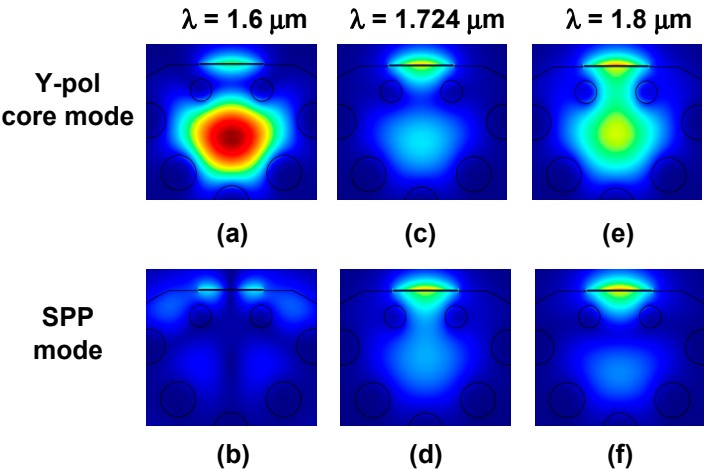

**Figure 3.** Light field distributions in the cross section of the MPF–SPR sensor for different wavelengths (**a**–**f**). The RI of the analyte is $n_a$ = 1.38.

### 3.2. Optimization of Air Holes' Diameter

The sensing performance of the SPR sensor depends on structural parameters. To determine a suitable fiber structure, the effects of various air holes' diameter values are first studied by adjusting the plasmon–wave excitation spectrum. We fix other structural parameters ($d_2$ = 1 μm, $t_{ITO}$ = 60 nm, h = 6.4 μm) and change the regular air holes' diameter from 1.6 to 2.0 μm. As shown in Figure 4a, with the increase in the size of regular air holes, the loss spectrum is red-shifted and flattened. More light energy is confined in the eccentric core with bigger outer air holes, so the resonance intensity is weakened, eventually leading to decreased loss peak values and increased FWHM. As shown in Figure 4b, the wavelength sensitivity can be considered to be almost stable under different regular air holes' diameter values. Here, we choose d = 1.6 μm.

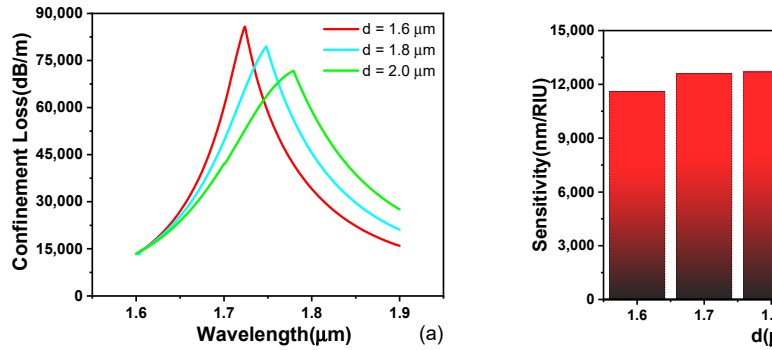

**Figure 4.** The loss spectra (**a**) and sensitivity (**b**) for various regular air holes' diameters d. The other parameters are $d_2$ = 1 μm, $t_{ITO}$ = 60 nm, h = 6.4 μm, and $n_a$ = 1.38.

Afterwards, we fix other structural parameters (d = 1.6 μm, $t_{ITO}$ = 60 nm, and h = 6.4 μm) and change the small air holes' diameter from 0.8 to 1.2 μm. As illustrated in Figure 5a, the confinement loss peak is blue-shifted slightly. When the small air holes' diameter equals to 1 μm, an extremely sharp spectrum appears. It has a small FWHM that can control and filtrate the spectral noise. Undersized air holes cause part of the light to leak out of the vertical channel. Oversized air holes have a great binding effect on coupling energy, resulting in an abnormal reduction in the loss peak. As illustrated in Figure 5b, the small air holes' diameter has little effect on the wavelength sensitivity, especially in the range of 0.9–1.1 μm. We can adjust the loss peak values by changing the small air holes' diameter. To observe the clearest loss spectrum, we choose $d_2$ = 1 μm. In conclusion, the sensing sensitivity does not change much with the sizes of any air hole. Our MPF–SPR

sensor has a big error tolerance, which reduces the accuracy requirements during fiber manufacturing and promotes its application in sensing and mass production.

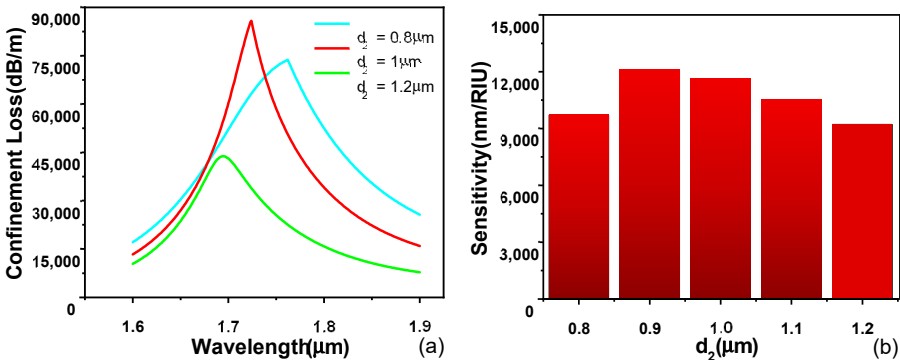

**Figure 5.** The loss spectra (**a**) and sensitivity (**b**) for various small air holes' diameters $d_2$. The other parameters are d = 1.6 μm, $t_{ITO}$ = 60 nm, h = 6.4 μm, $n_a$ = 1.38.

### 3.3. Optimization of ITO Thickness and Core-to-Surface Interval

The ITO film thickness and core-to-surface interval are also within the scope of our optimization. Figure 6 reveals influences of the ITO thickness $t_{ITO}$ on the sensing performance. When $t_{ITO}$ is varied from 50 to 70 nm, the confinement loss peak dramatically shifts to the long wavelength direction. At 60 nm, we see the sharpest loss peak and the highest loss peak value. This is the limit value of $t_{ITO}$, where the signal-to-noise ratio (SNR) is optimal in the sensing detection. ITO is a kind of exciting material, and its variation can cause an intense reaction of the SPR effect. The smaller thickness is unfavourable for the coupling of the y-polarized core mode and SPP mode. The larger thickness weakens the SPR effect because of the limitation of skin depth for the surface plasmon. It can be found that the wavelength sensitivity is improved as $t_{ITO}$ increases. Thus, $t_{ITO}$ = 60 nm is an optimum choice for our MPF–SPR sensor with a high SNR and wavelength sensitivity.

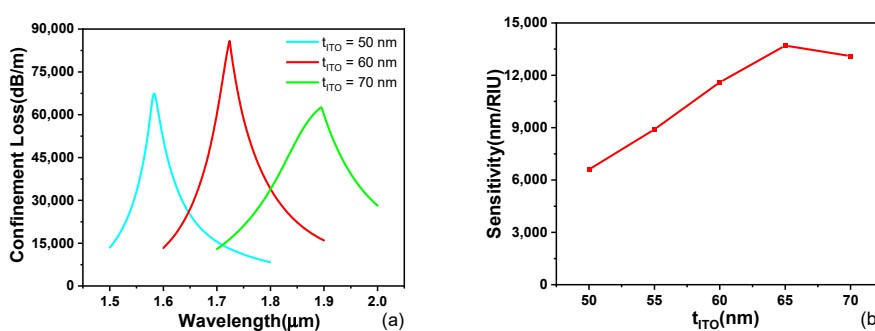

**Figure 6.** The loss spectrum (**a**) and sensitivity (**b**) for different ITO film thickness $t_{ITO}$. The remaining parameters are d = 1.6 μm, $d_2$ = 1 μm, h = 6.4 μm, and $n_a$ = 1.38.

Figure 7 reveals influences of the core-to-surface interval h on the sensing performance. The smaller core-to-surface interval h is the distance between the fiber core and the bottom of the ITO layer. When h is varied from 6.2 to 6.8 μm, the confinement loss peak shifts to the long wavelength direction and becomes observably flat. The shorter interval reduces the length of the leakage channel. It enhances the coupling of the y-polarized core mode and the SPP mode, which certainly increases the strength of loss. However, the lower grinding depth maintains a certain interval that can ensure the mechanical strength of the polished fiber. This is the particular advantage of our proposed sensor. It can be found that, as h increases, the wavelength sensitivity enhances constantly. Taking comprehensive consideration, h = 6.4 μm is a compromise choice for our MPF–SPR sensor.

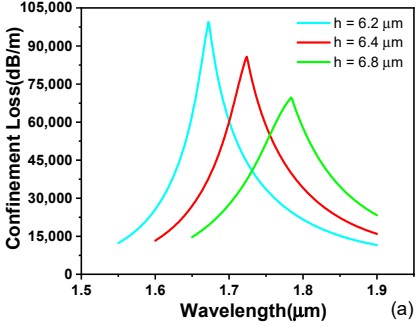 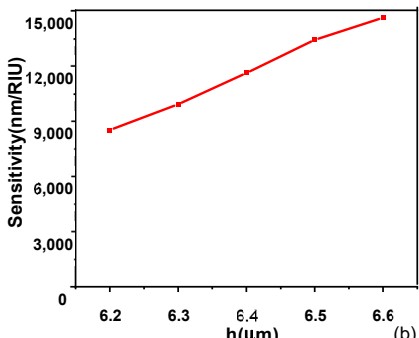

**Figure 7.** The loss spectrum (**a**) and sensitivity (**b**) for different core-to-surface interval h. The remaining parameters are d = 1.6 μm, $d_2$ = 1 μm, $t_{ITO}$ = 60 nm, and $n_a$ = 1.38.

### 3.4. Effects of Analyte RI Variation

The effects of analyte RI variation on the loss spectrum and sensitivity are displayed in Figure 8 and Table 1. The RI of analyte $n_a$ is from 1.32 to 1.39 with an interval of 0.01 RIU. The confinement loss peak is red-shifted with the increasing RI of the analyte. The higher RI reduces the RI difference between the eccentric core and analyte, and the light of longer wavelength can excite SPR effectively. When $n_a$ = 1.38, the confinement loss and wavelength sensitivity reach a maximum. Here, the best phase matching between the y-polarized core mode and SPP mode results in the strongest coupling directly. The polynomial fitting result of the resonance wavelength and analyte RI is displayed in Figure 8b. The result satisfies the numerical equation $y = 341.56x^2 - 244.58x + 163.25$. $R^2 = 0.9907$ means a highly fitting response of the polynomial curve. Our MPF–SPR sensor shows an excellent continuous response, and the detection system is accurate and reliable.

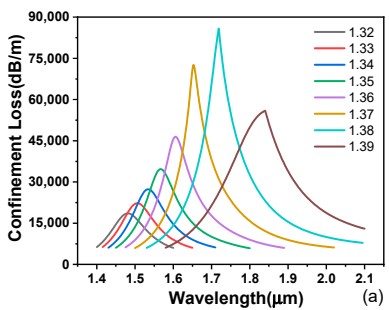 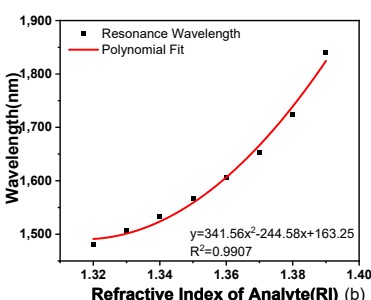

**Figure 8.** The loss spectra (**a**) and resonance wavelength (**b**) for various analyte's RI.

**Table 1.** Sensing performance of the MPF–SPR sensor.

| Analyte RI. | Resonance Wavelength (nm) | Peak Loss (dB/m) | Wavelength Sensitivity (nm/RIU) |
|---|---|---|---|
| 1.32 | 1481 | 18,564 | 2500 |
| 1.33 | 1506 | 22,283 | 2800 |
| 1.34 | 1534 | 27,350 | 3300 |
| 1.35 | 1567 | 34,677 | 3900 |
| 1.36 | 1606 | 46,446 | 4700 |
| 1.37 | 1653 | 72,614 | 7100 |
| 1.38 | 1724 | 85,819 | 11600 |
| 1.39 | 1840 | 55,957 | - |

Table 2 shows the characteristics of the proposed sensor compared with previous sensors. It indicates that the proposed sensor in this work has the advantage of higher refractive index detection. Meanwhile, the sensitivity remains higher among the previous sensors.

**Table 2.** Comparison of the proposed sensor with previous designs.

| Structure | RI Range | Max Sensitivity (nm/RIU) | Max Resolution | Year | Ref. |
|-----------|----------|--------------------------|----------------|------|------|
| D-shaped | 1.23–1.29 | 5500 | $7.69 \times 10^{-6}$ | 2017 | [29] |
| Dual-shaped | 1.27–1.32 | 13,500 | $7.41 \times 10^{-6}$ | 2018 | [18] |
| D-shaped | 1.20–1.29 | 11,055 | $9.05 \times 10^{-6}$ | 2019 | [17] |
| D-shaped | 1.19–1.29 | 10,700 | - | 2018 | [30] |
| D-shaped | 1.22–1.33 | 15,000 | $6.67 \times 10^{-6}$ | 2020 | [31] |
| D-shaped | 1.32–1.39 | 11,600 | - | 2022 | This work |

### 4. Conclusions

An ITO-coated MPF–SPR sensor was proposed for detecting the analyte RI. After the advantages of the design were summarized, the fiber structures were theoretically analysed. The simulation results indicated that the proposed sensor could achieve a maximum sensitivity of 11,600 nm/RIU in the RI range of 1.32 to 1.39. It was found that different structure parameters all have a certain influence on the confinement loss. Especially, we can achieve an extremely sharp peak with a suitable air holes' diameter and ITO film thickness. The wavelength sensitivity increases as the ITO film thickness and core-to-surface interval increase. However, the proposed sensor has a high fabrication tolerance and the air holes' size has little influence on the sensitivity. The simple design uses cheap sensing material, and the tiny grinding depth ensures the mechanical strength and complete air holes. Hence, our MPF–SPR sensor can provide satisfactory applications for biomedicine and chemistry.

**Author Contributions:** Conceptualization, J.C. and S.H.; methodology, Q.L.; validation, Q.L., J.C. and J.L.; formal analysis, J.C. and S.H.; investigation, J.C.; resources, J.C.; data curation, J.C.; writing—original draft preparation, J.L.; writing—review and editing, Q.L.; supervision, S.H.; project administration, J.L.; funding acquisition, S.H. All authors have read and agreed to the published version of the manuscript.

**Funding:** This research was funded by National Natural Science Foundation of China, grant number 61665005. The APC was funded by HongLiu First-Class Disciplines Development Program of Lanzhou University of Technology.

**Data Availability Statement:** Not applicable.

**Conflicts of Interest:** The authors declare no conflict of interest.

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
