# Peer review of "Investigation into Micro-Polishing Photonic Crystal Fibers for Surface Plasmon Resonance Sensing"

_crystals, doi:10.3390/cryst12081106_

Round 1
Reviewer 1 Report
The photonic-crystal-fiber plasmonic sensor is a popular subject of research and publications today. Thousands of papers were published last year, approximately 10% of which explicitly mention COMSOL as modelling tools. The D-shaped PCF, selected by the authors, is a popular design solution because it seemingly offers ease of evanescent field manipulation and support for the conducting layer, which is necessary for SPR guidance.
As such a structure is an obvious solution for plasmonic sensors, it has been extensively modelled and described in many of papers. With this being taken into account, the originality of the concept is minimal.
The authors do not try to investigate the technical difficulties that appear when such a sensor is put into production. Micropolishing is a good laboratory technique; however, low yields preclude large-scale industrial fabrication. ITO films, which the authors propose as a conductive layer, have nonuniform conductivity in their crosscection; this property, ignored in the paper, will influence the performance of the plasmonic sensor.
The title of the page should be changed to "Numerical investigation on micro-polishing photonic crystal fibers for surface plasmon resonance sensing" to reflect the actual content of the paper.
Author Response
Dear reviewer:
Thank you very much for your careful review and comments on our article “Investigation on micro-polishing photonic crystal fibers for surface plasmon resonance sensing” (Manuscript ID: crystals-1758756 ). Our revised manuscript is attached with the revision instruction. The reply and modification details of relevant questions are as follows:
Questions for us:
“The authors do not try to investigate the technical difficulties that appear when such a sensor is put into production. Micropolishing is a good laboratory technique; however, low yields preclude large-scale industrial fabrication. ITO films, which the authors propose as a conductive layer, have nonuniform conductivity in their crosscection; this property, ignored in the paper, will influence the performance of the plasmonic sensor. The title of the page should be changed to "Numerical investigation on micro-polishing photonic crystal fibers for surface plasmon resonance sensing" to reflect the actual content of the paper.”
Reply: At present, due to the limitations of experimental conditions, most of the exploration on the structure of PCF-SPR sensor is limited to theoretical simulation, and there are relatively few reports on experimental production. In terms of structure, D-type Micro-Polishing PCF-SPR has the greatest application potential and may be a better application direction in the future. Therefore, the research content of this article is mainly based on this idea. During the simulation of the designed sensor, the influence of the optical fiber structure and the thickness of the plated metal material on the sensor performance is discussed mainly under the condition that other parameters are ideal. Therefore, the influence of the conductivity difference caused by the uneven metal material in the cross section of the plated metal layer is ignored. At the same time, we will also seriously consider your proposal to change the title of the article to "Numerical investigation on micro-polishing photonic crystal fibers for surface plasmon resonance sensing".
Thank you again for your valuable suggestions!
Best wishes!
Your sincerely,
Qingmin Liu, Junpeng Chen, Shanglin Hou, Jingli Lei

Reviewer 2 Report
Manuscript Number: crystals-1758576
Authors: Qingmin Liu, Junpeng Chen, Shanglin Hou, Jingli Lei
Title: Investigation on micro-polishing photonic crystal fibers for surface plasmon resonance sensing
Reviewer’s comments: Major revision
In this manuscript, the authors propose and experimentally demonstrate an ITO-coated micro-polishing fiber (MPF)-based SPR sensor is proposed, consisting of 3 layers of regular air holes and 2 small symmetrical air holes. This proposed PCF-SPR sensor can achieve the measurement of refractive index by monitoring the wavelength sensitivity and the sharpness of loss peaks. Overall, the total manuscript is well-organized. It maybe published in the future when all glitches are fixed. Here are a few comments:
(1) There are a lot of grammatical errors in the manuscript, for example, the sentence in line 54. Besides, tense consistency should be seriously checked in the introduction section. The authors should check all these careless errors before the next submission.
(2) In the introduction, the authors give some discussion about limitation of several reported PSF-SPR sensors from the viewpoints of experiments or fabrication, however, the performance of this proposed sensor is just numerically analyzed and no experimental demonstration is provided, could please illustrate the advantages of the proposed sensor with comparison of them?
(3) It seems like an editing mistake in line 109 and line 128 with the number of 16 and 19, also in line 124-125 and line 130-131. In addition, these careless mistakes occurred too many times in the manuscript and they must be checked throughout and revised.
(4) The legend in Fig.2 is not clear. Also please give more explanation on the breakpoints of real part of RI at the wavelength of 1724 nm.
(5) FWHM is a critical factor of the proposed sensor, could you please discuss more on the figure of merits of the sensor? Or explain more on the relationship between FWHM and the sensitivity?
(6) May I know that it is a standard method for the geometric size determination and optimization or it is developed by the authors?
Author Response
Dear reviewer:
Thank you very much for your careful review and comments on our article “Investigation on micro-polishing photonic crystal fibers for surface plasmon resonance sensing” (Manuscript ID: crystals-1758756 ). Our revised manuscript is attached with the revision instruction. The reply and modification details of relevant questions are as follows:
Question 1: There are a lot of grammatical errors in the manuscript, for example, the sentence in line 54. Besides, tense consistency should be seriously checked in the introduction section. The authors should check all these careless errors before the next submission.
Reply 1: The grammar and tense of the manuscript have been checked in detail and modified accordingly.
Question 2: In the introduction, the authors give some discussion about limitation of several reported PSF-SPR sensors from the viewpoints of experiments or fabrication, however, the performance of this proposed sensor is just numerically analyzed and no experimental demonstration is provided, could please illustrate the advantages of the proposed sensor with comparison of them?
Reply 2: At present, due to the limitations of experimental conditions, most of the exploration on the structure of PCF-SPR sensor is limited to theoretical simulation, and there are relatively few reports on experimental production. In terms of structure, D-type Micro-Polishing PCF-SPR has the greatest application potential and may be a better application direction in the future. Therefore, the research content of this article is mainly based on this idea.
In the introduction, lines 52-76 of the article, we mentioned that compared with the analyte and metal coating difficulties of the metal inner coated sensor and the outer coated sensor, and the uncontrollable external disadvantages of the thickness and roughness of the metal film, the proposed micro-polishing D-type sensor can not only solve the difficulties of metal material coating, but also avoid the problem that the mechanical strength of most D-type PCF is reduced due to the large polishing depth in the production process.
Question 3: It seems like an editing mistake in line 109 and line 128 with the number of 16 and 19, also in line 124-125 and line 130-131. In addition, these careless mistakes occurred too many times in the manuscript and they must be checked throughout and revised.
Reply 3: The proposed editorial questions on lines 109 and 128, 124-125 and 130-131 have been revised in the article. In addition, the editing problems in other parts of the article have also been modified.
Question 4: The legend in Fig.2 is not clear. Also please give more explanation on the breakpoints of real part of RI at the wavelength of 1724 nm.
Reply 4: The problem of unclear legend in Figure 2 has been adjusted accordingly. As shown in Fig.2, the coupling resonance between the core mode and the SPP mode occurs at 1724 um, and then the two new modes are transmitted in the optical fiber with the increase of the incident wavelength. This phenomenon is called anti-cross effect, and the resonance position is called anti-cross point. When the incident wavelength is far shorter than the anti-crossing point, the energy of core mode is well-confined in the fiber core. As the wavelength increases, the coupling between SPP and core modes starts to appear. When wavelength reaches the anti-crossing point, we should notice that the electric field distribution of core and SPP modes are almost the same, and the Re(neff) of both two modes experienced a sudden change. As the wavelength continues to increase, the energy of core mode is thoroughly transferred into SPP mode, while the energy transition of SPP mode is quite the contrary.
Question 5: FWHM is a critical factor of the proposed sensor, could you please discuss more on the figure of merits of the sensor? Or explain more on the relationship between FWHM and the sensitivity?
Reply 5: In addition to sensitivity, another important parameter for evaluating sensor performance is the quality factor. It is defined by sensitivity and full width at half height:
Higher quality factor shows higher sensitivity and sharp loss spectrum. The sharp limit loss peak is very clear, and it is easy to detect the maximum point of the analyte loss spectrum. At the same time, it has a small full width at half height, which can control and filter the spectral noise. At the same time, the sensitivity also has a great relationship with the order of the incident light wave. The higher the order of the incident light wave mode, the greater the contribution to the sensitivity, but also weaken the quality factor. The more incident light modes participate in the sensing, the higher the sensitivity of the sensor will be. However, the quality factor will also drop sharply due to the broadening of the SPR formant. The main reason is that the more incident modes participate in the sensing, the more serious the mode superposition, which makes the SPR formant broaden more severely, and the worse the quality factor becomes.
Question 6: May I know that it is a standard method for the geometric size determination and optimization or it is developed by the authors?
Reply 6: The performance of the sensor is closely related to the size of the optical fiber and the sensing material. In the process of optimizing the sensor performance, the optical fiber size is mainly optimized by the finite element method to determine the optimal parameters. Therefore, the method used to determine and optimize its size is standard, but in the process of optimization, it is finally obtained through the author's continuous attempt to calculate.
Thank you again for your valuable suggestions!
Best wishes!
Your sincerely,
Qingmin Liu, Junpeng Chen, Shanglin Hou, Jingli Lei

Round 2
Reviewer 2 Report
The authors have addrerssed my questions well and revised the language errors, I think it can be accepted for publication in Crystal Journal